# Targeting Retinaldehyde Dehydrogenases to Enhance Temozolomide Therapy in Glioblastoma

**DOI:** 10.3390/ijms252111512

**Published:** 2024-10-26

**Authors:** Rafael Jiménez, Andrada Constantinescu, Muhube Yazir, Paula Alfonso-Triguero, Raquel Pequerul, Xavier Parés, Mileidys Pérez-Alea, Ana Paula Candiota, Jaume Farrés, Julia Lorenzo

**Affiliations:** 1Department of Biochemistry and Molecular Biology, Faculty of Biosciences, Universitat Autònoma de Barcelona, E-08193 Bellaterra, Spain; rafa95.jimenezaguilar@gmail.com (R.J.); paula.alfonso@icn2.cat (P.A.-T.); raquel.pequerul@uab.cat (R.P.); xavier.pares@uab.cat (X.P.); anapaula.candiota@uab.cat (A.P.C.); 2Institute for Biotechnology and Biomedicine, Universitat Autònoma de Barcelona, E-08193 Bellaterra, Spain; 3Unit of Research in Cellular and Molecular Biology, Advanced BioDesign, Saint-Priest, 69800 Lyon, France; andrada.constantinescu@a-biodesign.com (A.C.); muhube.yazir@a-biodesign.com (M.Y.); mileidys.perez@a-biodesign.com (M.P.-A.); 4Catalan Institute of Nanoscience and Nanotechnology (ICN2), CSIC and BIST, Campus UAB, E-08193 Bellaterra, Spain; 5Centro de Investigación Biomédica en Red de Bioingeniería, Biomateriales y Nanomedicina, Instituto de Salud Carlos III, E-08913 Bellaterra, Spain

**Keywords:** aldehyde dehydrogenase, cancer, enzyme inhibition, glioblastoma, retinoic acid

## Abstract

Glioblastoma (GB) is an aggressive malignant central nervous system tumor that is currently incurable. One of the main pitfalls of GB treatment is resistance to the chemotherapeutic standard of care, temozolomide (TMZ). The role of aldehyde dehydrogenases (ALDHs) in the glioma stem cell (GSC) subpopulation has been related to chemoresistance. ALDHs take part in processes such as cell proliferation, differentiation, invasiveness or metastasis and have been studied as pharmacological targets in cancer treatment. In the present work, three novel α,β-acetylenic amino thiolester compounds, with demonstrated efficacy as ALDH inhibitors, were tested in vitro on a panel of six human GB cell lines and one murine GB cell line. Firstly, the expression of the ALDH1A isoforms was assessed, and then inhibitors were tested for their cytotoxicity and their ability to inhibit cellular ALDH activity. Drug combination assays with TMZ were performed, as well as an assessment of the cell death mechanism and generation of ROS. A knockout of several ALDH genes was carried out in one of the human GB cell lines, allowing us to discuss their role in cell proliferation, migration capacity and resistance to treatment. Our results strongly suggest that ALDH inhibitors could be an interesting approach in the treatment of GB, with EC50 values in the order of micromolar, decreasing ALDH activity in GB cell lines to 40–50%.

## 1. Introduction

Gliomas are broadly classified as astrocytomas or oligodendrogliomas according to the predominant cell type observed on histological analysis [1]. Belonging to the former group of gliomas, glioblastoma (GB), which is classified as grade IV according to the WHO classification, is the most aggressive and prevalent of all the primary malignant central nervous system tumors [2]. It accounts for 60% of brain tumors in adults [3] and presents a dismal 5-year survival rate of only 5%. The median survival averages less than 15 months after diagnosis [4], which has remained nearly unchanged over the last 50 years [5].

The current standard of care for GB includes surgical resection of the tumor followed by postoperative adjuvant radiotherapy and chemotherapy [6], since this regimen was proven to have a survival benefit [7,8,9]. Chemotherapy is based on the administration of temozolomide (TMZ), an oral alkylating prodrug that elicits cytotoxicity by methylating guanine and adenine residues of DNA. This drug leads to lethal base pair mismatches that result in DNA strand breaks, inducing cell cycle arrest at the G2/M phase and eventually leading to cell apoptosis [10,11]. In addition, other authors suggested immune-related actions for TMZ beyond direct cytotoxicity [12]. Despite the treatment efforts, GB presents a formidable challenge due to its diffuse infiltration into the brain during the early stages of the disease, thus being nearly impossible to reach complete resections [13]. Moreover, remaining tumor cells often develop resistance to TMZ, leading to recurrence and tumor progression. This difficulty is the main pitfall of GB treatment and the reason why GB is still currently an incurable disease with a poor prognosis and few therapeutic advances over the past decades [14,15].

Understanding resistance to TMZ is not straightforward since it can be either inherently characteristic of certain tumors or acquired after initial treatment [16]. It is widely accepted that one of the primary contributors to TMZ resistance is the methylation status of the promoter region of the MGMT (O-6-methylguanine-DNA methyltransferase) gene. This gene regulates the expression of an endogenous DNA repair enzyme that directly counteracts DNA alkylation damage. Specifically, unmethylated tumors present higher levels of MGMT and commonly exhibit intrinsic resistance to TMZ and a worse clinical response in comparison to their methylated counterparts [17,18,19]. Besides MGMT and other DNA-repair systems, additional mechanisms contributing to TMZ resistance have been proposed over the last years, including autophagy, ferroptosis, alterations in key signaling pathways (Akt, Wnt/β-catenin, etc.) and, remarkably, the involvement of glioma stem cells (GSCs) [20,21,22]. GSCs constitute a small subset of undifferentiated cancer cells with inherent resistance to radiation and chemotherapy, and they also possess the unique ability to self-renew and maintain the tumor. GSCs are believed to activate after therapy and play a pivotal role in repopulating recurrent tumors [23]. Given the growing interest in specifically targeting GSCs to prevent tumor recurrence, several GSC markers have been established to better identify these cells. Among them, ALDH1A3, belonging to the aldehyde dehydrogenase (ALDH) superfamily of enzymes, is one of the most widely accepted and utilized biomarkers [24].

ALDHs catalyze the NAD(P)+-dependent oxidation of a wide range of endogenous and exogenous aldehydes to their corresponding carboxylic acids. Aldehydes are highly reactive molecules that can impair cellular function by forming adducts with DNA and proteins, so they are often cytotoxic and carcinogenic [25,26]. In this regard, ALDHs act as detoxifying enzymes, not only in normal cells, but also in cancer stem cells, where some enzyme isoforms were found overexpressed [27]. Specifically, cytosolic isoforms ALDH1A1, ALDH1A2 and ALDH1A3 play crucial roles in cell signaling via the oxidation of retinaldehyde to retinoic acid, which modulates a broad spectrum of biological processes including cell proliferation, differentiation, cell cycle arrest and apoptosis [28], processes that have been related to tumor initiation and progression. Other reported roles of ALDHs in cancer cells include the detoxification of antineoplastic drugs, resistance to radiation, involvement in migratory capacity, clonogenicity and metastatic potential. In addition, ALDHs are known to interact with several oncogenic pathways and their active participation in the immune microenvironment of cancer stem cells [29,30]. Given the multifaceted roles that they play, ALDHs are not solely regarded as consistent markers for identifying cancer stem cells but have also recently emerged as promising targets for pharmacological intervention.

In GB, the isoform ALDH1A3 emerges as a particularly relevant player. Several studies point out that ALDH1A3 is a key driver in the mesenchymal differentiation of GSCs [31,32]. Out of the different subtypes of GSCs established by omics analysis, the mesenchymal subtype is known to exhibit the most aggressive phenotype and is associated with diminished overall survival in GB patients [33], heightened tumor invasiveness [34], spreading and resistance to radiotherapy [35]. ALDH1A3 has proven to be the major contributor to ALDH activity in this subset of cells and to be responsible for their maintenance, self-renewal, survival, proliferation and tumorigenicity [36,37]. Furthermore, both ALDH1A3 and, to some extent, ALDH1A1 have been associated with the resistance of GB cells to TMZ treatment [38,39,40,41].

Thus, the search for isoform-selective ALDH inhibitors has emerged during the last years as an interesting approach for GB treatment. Recent studies demonstrated the efficacy of selective inhibitors and probes of ALDH1A3 against GB cell cultures and tumorspheres [42,43,44,45,46,47,48]. In the present work, we aim to contribute to this emerging and expanding field by investigating the effect of three novel ALDH inhibitors (namely, DIMATE and its analogues, ABD0099 and ABD0171) on in vitro cultured GB cell lines. Previous studies have already undertaken the kinetic characterization of these inhibitors against different ALDH isoforms, and some of them have shown promising results in addressing various cancer types, highlighting their potential for broader therapeutic applications [49,50,51,52,53,54]. DIMATE is currently being evaluated in clinical trials (Clinical trial NCT05601726). Our work seeks to explore the efficacy of ALDH inhibitors in GB, particularly when combined with TMZ, offering new insights into potential therapeutic strategies for this challenging disease.

## 2. Results

### 2.1. ALDH1A Isoforms Are Expressed Differently in a Panel of GB Cell Lines

The protein expression levels of ALDH isoforms ALDH1A1, ALDH1A2 and ALDH1A3 were assessed by capillary electrophoretic immunoassays using a comprehensive panel of seven GB cell lines, including six human cell lines (LN229, T98G, U251-MG, U373, U87-MG and A172) and one murine cell line (GL261). For comparative purposes, a non-cancerous human astrocytic cell line derived from the stem cell line Ax-0019 [55] was also included in this study.

In Figure 1, the protein expression profile of ALDH1A1, ALDH1A2 and ALDH1A3 is depicted for all the cell lines tested. Quantification of capillary-based immunoassay peaks is plotted as histograms in Appendix A. ALDH1A3 is highly expressed in all cell lines tested except for T98G. The observed low expression in T98G aligns with the findings reported by Wu et al. [56]. Conversely, ALDH1A1 and ALDH1A2 display the highest expression levels in LN229 and A172 cell lines. In the non-cancerous astrocytic cell line, included here as a control for comparative purposes with GB cell lines, ALDH1A2 and ALDH1A3 are strongly expressed, whereas ALDH1A1 is absent. It is important to note that this astrocytic cell line was derived from human-induced pluripotent stem cells, which may not faithfully recapitulate the real origin of astrocytes, and, thus, ALDH expression in this cell line may differ to some extent from that of real astrocytes. The results shown here indicate that the expression levels of ALDH1A3 are similar between the astrocytic cell line and the GB cell lines. However, more accentuated differences would be expected between astrocytes and subpopulations of GSCs, where ALDH1A3 is usually found overexpressed according to the literature. Such subpopulations differ in their molecular signatures and proliferative capacity, which allows them to adapt to therapeutic pressures and environmental changes, making treatment more challenging. This finding is supported by studies showing that different GSC subsets can exhibit unique molecular profiles, such as varying levels of ALDH activity, which are associated with resistance to standard treatments. Additionally, GSCs display a high degree of plasticity, meaning they can switch between different states ranging from stem-like to differentiated and the other way around, further contributing to tumor recurrence and therapeutic resistance. This variability complicates treatment as different subpopulations may respond differently to the same therapy [57].

### 2.2. ALDH Activity in A172 and GL261 Cells

Subsequent experiments primarily focused on the human GB A172 and the murine GB GL261 cell lines. The A172 cell line was selected due to its notably high protein expression of all three ALDH1A isoforms, as demonstrated in Figure 1 and Appendix A. GL261, the only murine cell line tested, was chosen because it is a suitable line to be used in immunocompetent models for future in vivo preclinical studies, as widely described by us and others in different therapeutic scenarios [58,59,60,61].

To evaluate the ALDH activity in A172 and GL261 cells, enzymatic activity assays were performed by using two different substrates: hexanal and all-*trans*-retinaldehyde. The former is the standard substrate routinely used for the kinetic characterization of ALDH inhibitors [62]; the latter is the physiological substrate of ALDH1A1, ALDH1A2 and ALDH1A3, which is oxidized to retinoic acid, which in turn regulates numerous signaling pathways related to cell proliferation and differentiation [28,44,63].

The absolute values of ALDH activity in A172 and GL261 cells, both with hexanal and all-*trans*-retinaldehyde as substrates, are shown in Table 1. Comparison of these values between cell lines indicates that ALDH activity is higher in A172 cells compared to GL261 cells with both substrates. Notably, retinaldehyde dehydrogenase activity was detectable only in A172 cells. This observation suggests that the ALDH1A content in the GL261 extract was insufficient to produce detectable levels of retinoic acid, consistent with the immunoblot analysis shown in Figure 1. Additionally, ALDH activity was higher using hexanal substrate than using retinaldehyde in both cell lines, probably because more ALDH isoforms are active with hexanal.

### 2.3. ALDH Knockout Impairs Growth Rate and Migration Capacity in A172 Cells and Enhances Sensitivity to the Standard Chemotherapeutic Agent TMZ

In order to gain more insight into the role of ALDH enzymes in GB, we decided to conduct knockout experiments in the human cell line A172 by using the CRISPR/Cas9 gene edition system, initially targeting *ALDH1A3*. To verify the effects of the knockout, we confirmed the absence of this enzyme at both the protein and RNA levels. Additionally, we extended the expression detection analysis to examine if there were any collateral effects upon eliminating *ALDH1A3*, such as overexpression of other subfamily members. From now on, the cell line A172 will be referred to as A172 WT, while the knockout cell line will be called A172 KO.

Firstly, RT-PCR and Western analyses were performed in order to validate ALDH as a target. As shown in Figure 2, the expression of several ALDHs was sharply reduced in A172 KO compared to A172 WT. At mRNA level (Figure 2B), all screened isoforms were found decreased, especially ALDH1A1 and ALDH1A3. At the protein level (Figure 2A), the most affected isoforms were the enzymes belonging to ALDH1 family (namely, ALDH1A1, ALDH1A2, along with ALDH1A3 and ALDH2). The expression of other isoforms, such as ALDH1B1 and ALDH3A1, was also decreased.

ALDH activity in extracts of A172 WT and A172 KO cells was then determined using hexanal and all-*trans*-retinaldehyde as routine and physiological substrates, respectively (Figure 3). As expected, enzymatic activity was much lower in A172 KO compared to A172 WT. In the case of hexanal substrate, a residual activity remained in A172 KO (Figure 3A), possibly due to the contribution of other ALDHs or aldehyde-metabolizing enzymes that were not affected by the knockout and might still be able to oxidize hexanal. Conversely, when all-*trans*-retinaldehyde was used as the substrate, activity was completely abolished in A172 KO (Figure 3B). This result was not surprising, since it is well known that the main ALDH isoforms responsible for retinaldehyde oxidation are those belonging to the ALDH1A subfamily, highly affected by the knockout.

Once ALDH expression and activity was checked in A172 WT and A172 KO cells, further experiments were performed to investigate the impact of the lack of ALDH activity on features such as cell proliferation and migration.

A growth curve was established for both A172 WT and A172 KO (Figure 4), which allowed us to detect a decrease in the growth rate at the exponential phase in the knockout cell line compared to the wild type. The doubling time, calculated between days 5 and 8 for A172 WT was 1.72 ± 0.38 days, whereas the doubling time for A172 KO was 2.27 ± 0.13 days. The slower growth rate in A172 KO cells may be attributed to the impairment of retinoic acid production due to the lack of ALDH1A enzymes, as this molecule is a well-known key regulator in the activation of a wide range of genes involved in cell proliferation. Similar results were obtained in other studies where ALDH knockout was carried out on GB cell lines, such as the one performed by Li et al. [44]. It is also conceivable that toxic aldehydes generated by cellular processes, such as lipid peroxidation, cannot be properly eliminated due to insufficient ALDH1 activity [40,51,52,53].

On the other hand, migration capacity was also impaired in A172 KO compared to A172 WT (Figure 5), with the percent of wound closure of the migration area after 30 h being 16% lower in knockout cells [53].

These results strongly suggest that ALDHs and their role in retinoic acid synthesis are important in cancer cell processes such as proliferation and migration, which are directly related to features such as invasiveness and aggressiveness in GB tumors.

The toxicity of all-*trans*-retinaldehyde, TMZ, and other FDA-approved drugs, namely cisplatin (CP), carmustine (BCNU) and etoposide (ETP), was compared between A172 WT and A172 KO (Table 2 and Figure 6) after 48 h of incubation. The sensitivity of A172 KO cells to all-*trans*-retinaldehyde increased by 25% with respect to that of A172 WT cells. This increase in sensitivity was expected since ALDH1A enzymes are mainly responsible for the metabolism of this molecule, which is highly toxic in the amounts tested [64,65,66]. Interestingly, the EC_50_ values of TMZ decreased in approximately that same proportion in A172 KO cells, which agrees with the results obtained by Wu et al. [40] and supports the idea that ALDHs are somehow involved in the resistance mechanism to TMZ in GB [38,39,40,41]. This result is also in good agreement with the synergistic effects observed during the combined treatment of ALDH1A inhibitors and TMZ (Figure 4), which reinforces the role of ALDH1A as a potential drug target. Among the other tested drugs, the percent reduction of the EC_50_ value in the KO cell line compared to the WT was the highest with CP. Interestingly, it was reported that the CP sensitivity of cancer cells can be potentiated [53] or restored in resistant cells [67] by ALDH inhibition. ALDH may exert its role in chemoresistance through the detoxification of aldehydes derived from ROS generation and lipid peroxidation.

### 2.4. ALDH Inhibitors Display Potent Cytotoxicity in the Low Micromolar Range in a Panel of GB Cell Lines, Surpassing the Efficacy of the Conventional Standard of Care, TMZ

The cytotoxicity of the gold standard for GB treatment TMZ and various ALDH inhibitors, (namely, DIMATE, ABD0099 and ABD0171) was assessed in the panel of GB cell lines. Cells were treated with each compound, and after 48 h treatments, the dose–response curves were plotted (Figure 7), and the EC_50_ values (Table 3) were calculated.

From the EC_50_ calculated values, it is clear that the ALDH inhibitors are significantly more cytotoxic compared to TMZ in all cell lines. Remarkably, there is a difference in approximately three orders of magnitude between the potency of ALDH inhibitors and TMZ, with one exception: the cell line U373, which seems to be more sensitive to TMZ than the others. It is worth noting that U373 is among the cell lines expressing lower levels of ALDH1A1 (Figure 1), and it was already reported that ALDH1A1 is a mediator of TMZ resistance [38]. However, it is important to highlight that, in our results, we did not observe a direct correlation between ALDH1A1 and ALDH1A3 protein expression levels, both involved in TMZ resistance according to the literature [38,40], and the observed effects of TMZ in this study, hinting that different factors could be behind such resistance in the complex in vivo environment.

The EC_50_ value of the ALDH inhibitors in all cell lines falls within the low micromolar range. The exception is the EC_50_ value of ABD0099 in the cell line T98G, which is 3- to 20-fold higher than in the rest of cases. A plausible explanation could be that this cell line exhibits comparatively lower expression levels of all ALDH1A investigated isoforms, as depicted in Figure 1.

### 2.5. ALDH Inhibitors Reduce the Enzymatic ALDH Activity in A172 and GL261 Cell Extracts

To evaluate the target engagement in cells, enzymatic activity assays were performed to assess the inhibitory capacity of DIMATE, ABD0099 and ABD0171 against cellular ALDH activity by using hexanal and all-*trans*-retinaldehyde. Such inhibition in GB cells could be of great interest for pharmacological purposes.

The remaining enzymatic activities of A172 and GL261 cell extracts were determined in the presence of each ALDH inhibitor. As shown in Table 4, ALDH activity with hexanal was decreased in the presence of a 15 µM inhibitor. This concentration is routinely used in single-point inhibition in vitro assays since it is appropriate for the initial assessment of inhibitory potency. ABD0171 seemed to be the most potent inhibitor of cellular ALDH activity both in A172 and GL261 cell lines since it displayed the lowest value of remaining activity.

Remaining activities with all-*trans*-retinaldehyde are shown in Table 5. In this case, ALDH activity in A172 was also decreased in the presence of inhibitors. Furthermore, the remaining activity decreased with increasing concentrations of inhibitors; thus, it is concentration-dependent. In this case, DIMATE seemed to be the most potent inhibitor. No results are shown here for GL261 cells since, as mentioned above, no retinaldehyde dehydrogenase activity could be detected in this cell line.

A graphical illustration of the inhibition assay with DIMATE using all-*trans*-retinaldehyde as a substrate is represented in Figure 8, where the chromatograms resulting from the HPLC analysis of retinoids are shown. As indicated, the peak corresponding to retinoic acid product decreased when a greater amount of DIMATE was present in the reaction mixture.

### 2.6. Combinations of TMZ with ALDH Inhibitors Act Synergistically to Cause Toxicity in GB Cell Lines A172 and GL261

In order to assess whether TMZ and ALDH inhibitors could act synergistically in the treatment of GB, drug combination assays were performed in A172 and GL261 cell lines.

The data analysis revealed that only the combination of TMZ with DIMATE was synergistic in the treatment of A172 cells (synergy score of 17.21), while the combination with other inhibitors resulted in an additive effect (Figure 9A). This result could be related to the fact that DIMATE was the best compound at inhibiting the production of retinoic acid (Table 5). This result suggests that ALDHs may be somehow implicated in the mechanism of resistance to TMZ in this cell line.

Conversely, combinations of TMZ with ABD0099 and ABD0171 were synergistic in GL261 (Figure 9B), suggesting that ALDHs may also be involved in the mechanism of resistance to TMZ in this cell line. Specifically, the most synergistic combination was that of TMZ with ABD0171 (synergy score of 17.62). Interestingly, previous results obtained by our research group indicated that ABD0171 was the most potent inhibitor towards ALDH1A3 (manuscript in preparation), which seems to be one of the predominant isoforms in GL261 (Figure 1). In fact, ABD0171 was the compound yielding the lowest value of remaining activity with hexanal in this cell line (Table 4).

Since the regions of synergy appear at mid to low concentrations of TMZ, combination therapies using ALDH inhibitors could potentially help reducing the amount of TMZ used compared to TMZ monotherapy.

### 2.7. TMZ and ALDH Inhibitors Trigger Apoptosis and Induce ROS Generation

In order to further investigate how the ALDH inhibitors tested in this study elicit their toxicity on GB cell lines, the cell death mechanism and generation of ROS were investigated in A172 and GL261 cell lines. Moreover, TMZ was also included in these tests.

A cell death analysis by flow cytometry after a 24 h treatment with the specified compounds indicated that there is a trend towards apoptosis, rather than necrosis, since the number of cells labeled by annexin V increased to a greater extent than those labeled by propidium iodide when compared to untreated cells, in both A172 and GL261 cell lines, as shown in Figure 10. In fact, a statistical analysis revealed that only changes in apoptotic cells, and not in necrotic cells, were significant compared to the respective untreated controls.

The process by which cell death is triggered in A172 cells appears to be independent of ROS-induced damage, as none of the compounds generated significant ROS levels compared to untreated cells under the experimental conditions tested (Figure 11). In contrast, in GL261 cells, death might be induced at least in part by ROS, as treated cells showed slightly elevated ROS levels compared to the control. Notably, GL261 cells treated with compounds ABD0099 and ABD0171 displayed higher ROS levels than cells treated with DIMATE. Interestingly, TMZ combinations with ABD0099 and ABD0171 were more synergistic than that with DIMATE in this cell line (Figure 9), suggesting that ROS could be somehow involved in the mechanism of synergy.

## 3. Discussion

There is mounting evidence of several ALDH isoforms being relevant players and biomarkers in GB. Among these, ALDHA1 and ALDH1A3 have been associated with distinct molecular subtypes of GB and GSCs. Notably, ALDH1A3, rather than ALDH1A1, is predominantly expressed in GSCs, whereas non-stem glioma cells comparably express both isoforms [68]. In fact, several studies indicate that ALDH1A3 is associated with the mesenchymal subtype of GSCs and that retinoic acid and glycolytic metabolism are involved [31,32,69,70]. ALDH1A3 is the major contributor to ALDH activity in GSCs and responsible for their maintenance, self-renewal, survival, proliferation and tumorigenicity [31,36,37,56,70,71,72]. ALDH1A2 was expressed in the GB microenvironment, especially in M2 GB-associated macrophages [73].

The regulation of ALDH1A3 expression at epigenetic and protein level correlates with GB prognosis and treatment. Thus, the hypermethylation status of the *ALDH1A3* gene promoter in GB patients predicted a better prognosis with an accompanied low level of the ALDH1A3 protein [74]. Additionally, the post-transcriptional regulation of ALDH1A3 by proteasomal degradation was reported in GB cell lines. ALDH1A3 was regulated by autophagy during high concentration-TMZ treatment, highlighting the role of this enzyme in chemoresistance [39].

Both ALDH1A3 and, to some extent, ALDH1A1 have been associated with the resistance of GB cells to TMZ treatment [38,39,40]. ALDH1A3 knockout GB cells were more sensitive to TMZ, and this process was mediated by increased oxidative stress, which led to lipid peroxidation, yielding active aldehydes that could not be detoxified by ALDH enzymatic activity [40].

Validation of ALDH as a pharmaceutical target was performed by genetic silencing using the CRISPR/Cas9 KO technology in the A172 cell line. Our focus on silencing ALDH1A3 was motivated not only because it was abundantly expressed in GB cells but also because of its important role in GB biology. Several ALDH genes resulted as downregulated at mRNA, protein and enzymatic activity levels (Figure 2 and Figure 3). At this time, the mechanism by which the ALDH1A3 CRISPR/Cas9 KO plasmid also affected the expression of other ALDH genes is unclear to us. In these cells, ALDH1A3 might be particularly relevant orchestrating different biological processes, either energetic or metabolic processes, for example, and its knockout seems to have a profound regulatory effect on transcription and/or protein expression levels of at least few other ALDHs. This effect might have gone unnoticed in previous ALDH1A3 gene knockout experiments due to the focus on ALDH1A3 alone [39,44,75,76]. In a recent study, however, the genetic silencing of ALDH1A3 induced the downregulation of ALDH1A1, which may be related to transcriptional feedback regulation of some ALDH genes by retinoic acid [77,78]. The fact that retinaldehyde dehydrogenase activity (all-*trans*-retinoic acid production), mainly due to ALDH1A isoforms, was completely abolished in the A172 KO cell line would support this hypothesis. More importantly, knockout experiments involving several ALDHs in A172 cells revealed reduced growth rates and diminished migration capacity and also sensitized them to all-*trans*-retinaldehyde and some FDA-approved drugs, such as TMZ and CP. Since the production of retinoic acid from retinaldehyde was indeed impaired in this knockout cell line, this outcome illustrates the importance of ALDH1A isoforms and retinoic acid production in maintaining the tumor phenotype of A172 WT cells. These results strongly suggest that ALDHs and their role in retinoic acid synthesis are important in cancer cell processes such as proliferation and migration, directly related to features such as invasiveness and aggressiveness in GB tumors. Our ALDH1A3 knockout results in A172 glioblastoma cells align with other studies showing that reduced ALDH activity increases sensitivity to chemotherapeutic agents like TMZ, possibly due to enhanced lipid peroxidation and impaired autophagy. Similar to findings by Wu et al. [40,79] and Li et al. [80], the knockout disrupted retinoic acid biosynthesis, leading to reduced cell proliferation and migration. This result supports the potential of targeting ALDH1A3 as a therapeutic strategy in glioblastoma.

Having validated ALDH as a drug target, the use of ALDH inhibitors is granted as an alternative approach for GB treatment. Pan-ALDH inhibitors, such as disulfiram, DEAB or gossypol, were used, resulting in cytotoxic effect, particularly with TMZ [32,38,46,48,81,82]. ALDH1A3-selective inhibitors also demonstrated some efficacy against GB cell monolayers and tumorspheres, further supporting their potential in GB treatment [42,43,44,46,47,72]. In the present work, three novel ALDH inhibitors, namely, DIMATE and two of its analogues, were tested as promising chemotherapeutic candidates for the treatment of GB. We confirmed that ALDH inhibitors reduced the enzymatic ALDH activity in both A172 and GL261 cell extracts. These inhibitors were much more cytotoxic than the standard of care TMZ in a panel of seven GB cell lines. Although the EC_50_ values for TMZ in the studied cell lines agree with those reported by other authors in GL261 cells [12,83], they are higher than those for human GB cell lines [84,85,86,87,88]. It is conceivable that the various experimental approaches (i.e., the seeding ratio for cells, time of drug exposure or cell viability assay) could account for the differences observed. Despite this observation, the reported EC_50_ values for TMZ were at least two orders of magnitude higher than those for DIMATE and its analogues. Interestingly, some combinations of these ALDH inhibitors showed synergy with mid to low concentrations of TMZ, meaning that combination therapies could potentially help reducing the therapeutic dose administered, compared to TMZ monotherapy.

In a similar study in lung adenocarcinoma cell lines, Rebollido-Rios et al. [53] found that DIMATE induced apoptotic cell death by causing the intracellular accumulation of apoptogenic aldehydes hydroxynonenal (HNE) and malondialdehyde (MDA), which led to high levels of ROS and a drop in the levels of GSH. GSH, a key molecule for the detoxification of ROS in cells [89], was shown to play a crucial role in cell survival against DIMATE-induced apoptosis. A feasible explanation for the fact that cells treated with ALDH inhibitors show higher levels of ROS is that ROS generated by cellular processes such as lipid peroxidation are no longer detoxified by ALDHs, hence their levels increase in the cell. In the present work, however, ALDH inhibitors tested did not generate a significant increase in ROS levels in A172 cells, but this was not the case in GL261 cells, and the reason might be the level of ALDH activity present in each cell line. As shown in Figure 1, A172 expresses quite higher levels of ALDH enzymes than GL261. Accordingly, A172 cells naturally exhibit higher ALDH activity compared to GL261 cells (Table 2). This activity could explain that GL261 cells are more prone to oxidative stress than A172, as clearly suggested by the higher levels of ROS observed when cells are treated with H_2_O_2_. And yet, it is unclear whether different amounts of ALDH inhibitors might be needed in each case to abolish cellular ALDH activity, which merits further investigation.

Limitations of the current study while translating into in vivo applications include the following: authors are aware that unraveling the in vitro pathways and mechanisms is necessary, but not sufficient, to ensure successful translation. The considerable heterogeneity of GB tumors, treatment-derived mutations and microenvironment changes cannot be properly mimicked in vitro and should be carefully evaluated. Although it is unclear whether the results with the chosen cell lines could be fully extrapolated to other models and conditions, which is still to be confirmed, we consider that the results clearly warrant a preclinical in vivo investigation, in which a more realistic tumor microenvironment is present. This step, combined with a predefined chosen timepoint validation of ALDH local action in preclinical tumors, will pave the way for the final translational step. At present, some clinical trials have addressed the use of ALDH inhibitors in glioblastoma treatment, mostly centered in the use of the Disulfiram/Copper combination with TMZ (NCT01777919, NCT01907165, NCT02715609). Some concerns arose about dose-limiting toxicities, and modest results were reported, which warrants further investigation regarding schedule protocols and doses needed in the human scenario, as well as novel formulations such as nanoencapsulation. Other approaches such as CUSP9 [90] (clinical trial NCT02770378) investigated the promising combined use of several repurposed drugs in GB, including Disulfiram, highlighting the benefits of a multiapproach therapeutic protocol in GB. The differential expression of ALDH in stem-like cells was also tackled in therapy with approaches different from ALDH inhibition, such as the use of nanodisc vaccination against ALDH combined with immune checkpoint therapy [91].

Finally, we should not neglect the role of ALDHs in other cell types present in the tumor microenvironment, such as macrophages (e.g., ALDH1A2 is associated with M2 macrophages and may represent a marker for a subset of such cell types, which have a known protumoral effect [73]), hinting at an additional beneficial effect of ALDH inhibition in GB treatment.

## 4. Materials and Methods

### 4.1. Cell Lines and Cell Culture

All human GB cell lines (LN229 CVCL_0393, T98G CVCL_0556, U251-MG CVCL_0021, U373 CVCL_2219, U87-MG CVCL_0022 and A172 CVCL_0131) were acquired from ATCC and cultured in a DMEM medium (Life Technologies, Carlsbad, CA, USA) with a 10% FBS (Life Technologies, Carlsbad, CA, USA) without antibiotics. Cells were incubated in a humidified atmosphere with 5% CO_2_ at 37 °C. Murine cell line GL261 CVCL_Y003 was acquired from DSMZ (German Collection of Microorganisms and Cell Cultures GmbH, Leibniz, Germany) and cultured in the same conditions as the human GB cell lines. The non-cancerous human astrocytic cell line Ax-0019, obtained as described in [55], was kindly provided by Dr. Arranz’s research group (Laboratory of Humanized Models of Disease, Achucarro Basque Center for Neuroscience, Science Park of the UPV/EHU, Leioa, Spain). CVCL corresponds to the cellosaurus identification of each cell line.

### 4.2. Compounds (ALDH Inhibitors)

The ALDH inhibitors tested in this study, namely DIMATE, ABD0099 and ABD0171, were provided by Advanced BioDesign (Lyon, France). Their structures are depicted in Figure 12.

### 4.3. Western Blot Analyses

Automated capillary Western analysis was performed at Advanced BioDesign, on a WES system (ProteinSimple, San Jose, CA, USA) [92] using a 12–230 kDa Separation Module (ProteinSimple, SM-W004) and the Assay Module Anti-rabbit HRP Detection Module (ProteinSimple, DM-001), according to the manufacturer’s instructions. Samples were diluted at 1 mg/mL in Sample Buffer 0.1X (10X Buffer provided in the Separation Module), then mixed with Fluorescent Master Mix, vortexed and heated 5 min at 95 °C, then kept on ice. The samples, blocking reagent (antibody diluent), primary antibodies (in antibody diluent), HRP-conjugated secondary antibodies and chemiluminescent substrate were pipetted into the plate (provided in the Separation Module). The following instrument settings were used: stacking and separation at 375 V for 25 min; blocking reagent for 5 min; both the primary and secondary antibody for 30 min (except for 1 h of incubation in the case of the ALDH1A2 primary antibody); luminol/peroxide chemiluminescence detection for approximately 15 min (exposures of 1-2-4-8-16-32-64-128-512 s). The resulting graphs were checked, and the automatic peak detection was manually corrected. Data were visualized as lanes. Table 6 below describes the characteristics of antibodies used in the immunoassay, purchased from Proteintech (Rosemont, IL, USA), VWR (Radnor, PA, USA) and Abcam (Cambridge, UK).

### 4.4. Cell Viability Assays

Cells were seeded in 96-well plates at a density of 2000 cells per well for all human GB cell lines, and 4000 cells per well for GL261, and were incubated overnight at 37 °C and 5% CO_2_. The following day, cells were treated with different concentrations of the compound of interest and incubated in the conditions mentioned above for 48 h. Cell viability was measured by PrestoBlueTM assay (ThermoFisher, Waltham, MA, USA). The reagent was added to each well at 10% *v*/*v*, and fluorescence was read after 3 h of incubation in a SPARK multilabel plate reader, using an excitation wavelength at 531 nm and an emission wavelength at 572 nm.

Cell viability assays using single compounds: these assays were performed with ALDH inhibitors at concentrations ranging from 0 to 1 mM; TMZ (Merck, Darmstadt, Germany) at concentrations ranging from 0 to 20 mM; all-*trans*-retinaldehyde (Merck, Darmstadt, Germany) at concentrations ranging from 0 to 20 µM; CP (Merck, Darmstadt, Germany) at concentrations ranging from 0 to 1 mM; BCNU (Merck, Darmstadt, Germany) at concentrations ranging from 0 to 750 µM; and ETP (Merck, Darmstadt, Germany) at concentrations ranging from 0 to 250 µM. The proportion of ethanol or DMSO solvents never exceeded 1% *v*/*v* in the final volume of the well. In the case of all-*trans*-retinaldehyde, the compound was added to the wells under dim red light in order to prevent the photoisomerization of retinoid double bonds. EC_50_ values were calculated by nonlinear fitting of the obtained data to a sigmoidal plot using GraFit 5.0 (Erithacus software, http://www.erithacus.com/, accessed on 26 August 2024), with the following 4-parameter Equation (1):(1)$$y=\frac{range}{1+{\left(\frac{x}{{EC}_{50}}\right)}^{s}}+background$$
where “*y*” is the percentage of viable cells, “*x*” is the concentration of the compound, “*background*” is the minimum y value, range is the fitted uninhibited value minus the background and “*s*” is a slope factor.

Synergy assays: ALDH inhibitors and TMZ were concomitantly added to the wells in the concentrations mentioned above. Data resulting from these assays were analyzed with SynergyFinder software version 3.0 [93] (https://synergyfinder.fimm.fi/, accessed on 26 August 2024) in order to obtain the synergy maps and scores, calculated in accordance with the reference model developed by Bliss [94].

### 4.5. ALDH Activity Assays

ALDH activity assays were performed with cells collected during the exponential phase of growth. First, cell lysates were obtained using the M-PER reagent (ThermoFisher, Waltham, MA, USA), following the manufacturer’s instructions. Subsequently, total protein concentration of the lysate was determined by performing a Bradford assay.

Activity assays using hexanal as a substrate: Enzymatic activity was monitored using a Cary Eclipse (Varian, Palo Alto, CA, USA) fluorimeter at 37 °C. All reactions were performed in quartz cuvettes in a final volume of 1 mL, using 50 mM HEPES (Merck, Darmstadt, Germany), 50 mM of MgCl_2_, pH 7.2 as the reaction buffer, in the presence of 1% ethanol (ethanol is the solvent in which ALDH inhibitors were diluted). After adding the cell lysate and a 15 µM inhibitor to the reaction buffer, the mixture was incubated for 20 min at 37 °C to let the inhibitor bind to the enzyme. Then, a 0.5 mM NAD^+^ cofactor (Apollo Scientific, Bredbury, UK), a 5 µM NADH (Apollo Scientific, Bredbury, UK) and, finally, a 250 µM hexanal substrate (Merck, Darmstadt, Germany) were added to start the reaction. Fluorescence of NADH was followed at 460 nm with excitation at 340 nm and a spectral bandwidth of 10 nm. Five µM NADH was added to the reaction mixture as an internal standard to obtain absolute reaction rates, which were calculated according to the Equation (2):(2)$$v=\frac{dF}{dt}\cdot \frac{C_{st}}{F_{st}}$$
where “*C_st_*” is the standard NADH concentration, “*F_st_*” is the standard fluorescence and “*dF*/*dt*” is the slope of the time-dependent fluorescence [95].

Specific activity was expressed in milliunits (mU) per mg of total protein of the lysate, 1 mU defined as 1 nmol of product formed per min. Percentages of remaining activity were calculated for each reaction containing an inhibitor, relative to a control in absence of an inhibitor.

For statistical analysis, unpaired *t* tests with Welch’s correction were performed. Statistical significance was set at *p* values below 0.05.

Activity assays using all-*trans*-retinaldehyde as a substrate: Enzymatic activity was determined using an HPLC-based method. All reactions were performed in glass disposable tubes in a final volume of 0.5 mL, in DMEM (Life Technologies, Carlsbad, CA, USA) with a 10% FBS (Life Technologies, Carlsbad, CA, USA), in the presence of 1% ethanol. First, the culture medium, cell lysate (at a final total protein concentration ranging from 1 to 2 mg/mL) and the inhibitor (at final concentrations of 5, 50 and 250 µM) were incubated for 1 h at 37 °C to let the inhibitor bind to the enzyme. Then, a 0.5 mM NAD^+^ cofactor (Apollo Scientific, Bredbury, UK) and a 10 µM all-*trans*-retinaldehyde substrate (Merck, Darmstadt, DE) were added to start the reaction. From the moment the substrate was added to the reaction mixture, the experiment was carried out under dim red light in order to prevent the photoisomerization of retinoid double bonds. After 1 h at 37 °C, the reaction was stopped by adding 1 mL of cold methanol. Then, 0.1 mL of 2.5 M ammonium acetate, pH 4.5, was added in order to acidify the aqueous phase and facilitate the retinoic acid recovery [96]. Subsequently, retinoid extraction was performed by two rounds of addition of 2 mL of hexane, vortex mixing and centrifugation at 16,110× *g* for 1 min. From here, sample treatment, HPLC, retinoid quantification and activity calculation were carried out as previously described [63,64]. Specific activity was expressed in milliunits (mU) per mg of total protein of the lysate, 1 mU defined as 1 nmol of product formed per min. Percentages of the remaining activity were calculated for each reaction containing an inhibitor, relative to a control in absence of an inhibitor.

For statistical analysis, unpaired *t* tests with Welch’s correction were performed. Statistical significance was set at *p* values below 0.05.

### 4.6. Cell Death Analyses

Cells were seeded in 12-well plates at a density of 16,000 cells per well for A172 and at a density of 32,000 cells per well for GL261 and incubated overnight at 37 °C and 5% CO_2_. The following day, cells were treated with the required concentration of ALDH inhibitors (5 µM in all cases except for 20 µM of ABD0099 in GL261) or TMZ (5 mM) and incubated in the conditions mentioned above for 24 h. After this incubation time, cells were collected and treated with the eBioscience^TM^ Annexin V-FITC Apoptosis Detection Kit (ThermoFisher, Waltham, MA, USA), following the manufacturer’s recommendations. Cell samples were analyzed in a Cytoflex LX flow cytometer (Beckman Coulter, Brea, CA, USA), and the cell death mechanism was assessed by calculating the percentage of cells marked with annexin V (marker of apoptosis) or propidium iodide (indicator of necrosis), and comparing it to the corresponding untreated control.

For statistical analysis, unpaired *t* tests with Welch’s correction were performed. Statistical significance was set at *p* values below 0.05.

### 4.7. ROS Production Analyses

Cells were seeded in 96-well plates at a density of 20,000 cells per well and incubated overnight at 37 °C and 5% CO_2_. The following day, cells were treated with 25 µM of 2′,7′-dichlorofluorescin diacetate (DCFDA) (Merck, Darmstadt, Germany) and incubated in the dark under the conditions mentioned above for at least 30 min. Then, DCFDA was removed from the wells, and cells were treated with ALDH inhibitors or TMZ at the EC_50_ concentration. Cells were incubated for 6 h, and the fluorescence was read in a SPARK multilabel plate reader, using an excitation wavelength at 485 nm and an emission wavelength at 535 nm.

For statistical analysis, unpaired *t* tests with Welch’s correction were performed. Statistical significance was set at *p* values below 0.05.

### 4.8. CRISPR/Cas9 Knockout

A172 cells were transfected with a CRISPR/Cas9 plasmid (Santa Cruz Biotechnology, Dallas, TX, USA, SC-401630: ALDH1A3 CRISPR/Cas9 KO plasmid (h)) following manufacturer’s instructions. Forty-eight hours after transfection, single transfected cells were sorted in a 96-well plate using a BD FACSJazz Cell Sorter (BD Biosciences, Franklin Lakes, NJ, USA). Clones were allowed to grow at 37 °C and 5% CO_2_ for weeks and then scaled up to T75 flasks for further use. Although several clones were able to grow, only one of them was selected for all the experiments performed on the KO cell line.

In parallel, a negative control obtained from the transfection of A172 cells with a control plasmid (Santa Cruz Biotechnology, Dallas, TX, USA, SC-418922: control CRISPR/Cas9 plasmid) was also included in this study.

### 4.9. RT/PCR Analyses

RNA from A172 WT and A172 KO cells was first obtained using the E.Z.N.A. Total RNA Kit 1 (Omega bio-tek, GA, USA), following the manufacturer’s instructions. Extracted RNA was stored at −80 °C until use. Next, cDNA was obtained from RNA using the qScript XLT cDNA SuperMix (Quantabio, Qiagen Beverly, MA, USA), following the manufacturer’s instructions. Finally, the PCR reaction was performed with a 1 µg template cDNA using the PerfeCTa SYBR Green FastMix, ROX (Quantabio, Qiagen Beverly, MA, USA), following the manufacturer’s instructions. The primers used for each ALDH isoform are indicated in Table 7.

The relative mRNA expression of each ALDH isoform in A172 WT and A172 KO cells was quantified by the delta–delta Ct method, using the housekeeping gene of β-actin as a reference for normalization.

### 4.10. Cell Growth Curve

A172 WT and A172 KO cells were seeded in a total of 36 culture dishes with a 10 cm diameter at a density of 300,000 cells per dish. During the following 12 days, cells from three different dishes for each cell line were counted every 24 h using Trypan Blue staining. The doubling time (DT) for each cell line was calculated according to Equation (3) (as detailed in the ATCC Animal Cell Culture Guide, https://www.atcc.org/resources/culture-guides/animal-cell-culture-guide, accessed on 26 August 2024):(3)$$DT=T\cdot \frac{\ln2}{\ln\left(\frac{X_{e}}{X_{b}}\right)}$$
where “*T*” is the incubation time, “*X_b_*” is the cell number at the beginning of the incubation time and “*X_e_*” is the cell number at the end of the incubation time. The *DT* of each cell line was calculated by taking two points from the exponential phase of the growth curve, specifically day 5 (at the start of the exponential phase, where the number of cells in each cell line was still very similar) and day 8 (a day within the exponential phase).

For statistical analysis, bilateral comparisons (Student’s *t*-test) were performed. Differences were considered significant when *p* < 0.05.

### 4.11. Migration Assays

Migration assays were performed at Advanced BioDesign, using the Oris^TM^ Cell Migration Assembly Kit–FLEX (Platypus Technologies, Madison, WI, USA). First, the wells of a 96-well black plate with a clear bottom were coated with 20 µg/mL bovine fibronectin (PromoCell, Heidelberg, Germany) to avoid cell detachment in following steps of the protocol. The plate was incubated for 1 h at room temperature, protected from light. Then, the excess fibronectin was removed, Oris^TM^ Cell Seeding Stoppers were inserted into the desired wells, and 50,000 cells/well were seeded in a final volume of 100 µL per well. The seeded plate was incubated overnight at 37 °C to allow cells to adhere. The following day, the Oris^TM^ Cell Seeding Stoppers were removed, the wells were washed with PBS, and calcein dye (ThermoFisher, Waltham, MA, USA) (diluted in culture medium without FBS, to a final concentration of 0.25 µM) was added to the wells. The plate with calcein was incubated for 20 min at 37 °C. After that step, calcein was removed, the wells were washed with PBS and a complete culture medium was added to the wells. Fluorescence images at time 0 and after 30 h were obtained in a SpectraMax plate reader (Molecular Devices, San Jose, CA, USA), using an excitation wavelength of 456 nm and an emission wavelength of 541 nm. Migration areas were analyzed by ImageJ software version 1.53a (https://imagej.net/ij/, accessed on 26 August 2024), and migration was determined as percent closure calculated as per Equation (4):(4)$$Percent\;closure=\frac{premigration\;area-migration\;area}{premigration\;area}\cdot 100$$
where premigration area is the area at time 0 h.

For statistical analysis, unpaired *t* tests with Welch’s correction were performed. Statistical significance was set at *p* values below 0.05.

## 5. Conclusions

In this work, knockout experiments involving several ALDHs in A172 cells revealed impaired all-*trans*-retinaldehyde activity, reduced growth rates, diminished migration capacity and increased sensitivity to all-*trans*-retinaldehyde and some FDA-approved drugs, such as TMZ and CP. Along with synergy experiments, these findings might indicate that some retinoic acid-activated pathways could be involved in the TMZ resistance mechanism in GB. Taken together, these results suggest that ALDH inhibition could be a promising strategy in the treatment of GB. In this regard, three novel ALDH inhibitors, namely, DIMATE, ABD0099 and ABD0171, were studied as promising chemotherapeutic candidates for the treatment of GB. These inhibitors demonstrated significantly higher cytotoxicity compared to the standard of care drug TMZ across a panel of seven GB cell lines. In addition, these compounds were able to reduce the ALDH activity in A172 and GL261 cell extracts, using either hexanal or the physiological substrate all-*trans*-retinaldehyde. Notably, these compounds exhibited synergistic effects when combined with TMZ. Furthermore, our data suggest that the ALDH inhibitors induce apoptosis rather than necrosis in treated cells. Specifically, treated GL261 cells displayed slightly higher levels of ROS, which could be related to the mechanism of death, although further experiments are needed to confirm this hypothesis. Overall, our findings support the potential of ALDH inhibitors, particularly in combination with TMZ, to improve therapeutic outcomes in GB. The positive results from these novel compounds warrant further investigation in preclinical and clinical settings to fully elucidate their therapeutic potential and optimize GB treatment strategies.

## 6. Patents

MPA is an inventor on a patent related to the synthesis and use of ALDH inhibitors discussed in this article.

## Figures and Tables

**Figure 1 ijms-25-11512-f001:**
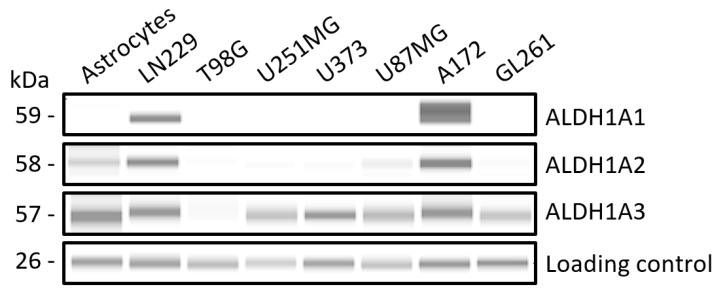
Western analysis of ALDH1A isoform expression in a panel of GB cell lines and astrocytes. Comparisons can be made across all cell lines for a single isoform, but direct comparisons between different isoforms within the same cell line are not possible due to antibody affinity differences and consequent variations in experimental conditions. The image was cropped to show the signals corresponding to ALDHs only.

**Figure 2 ijms-25-11512-f002:**
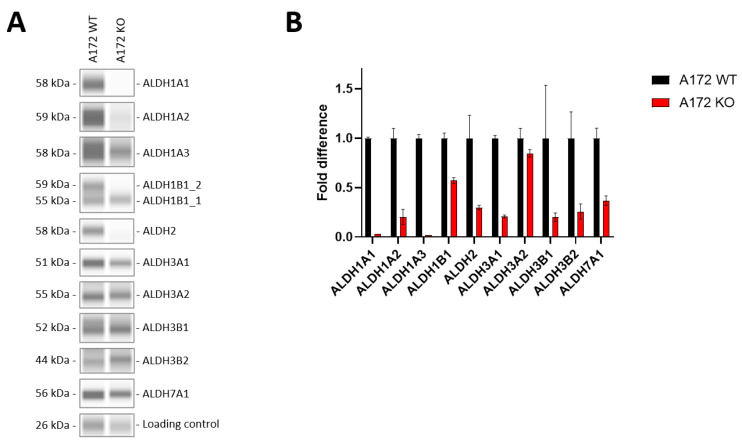
Validation of ALDH knockout in the human GB cell line A172. (**A**) Western analysis of ALDH expression at the protein level in A172 WT and A172 KO; (**B**) RT-PCR analysis of ALDH expression at mRNA level in A172 WT and A172 KO. Data are the mean ± SD of triplicates (n = 3); values were normalized by the expression of the housekeeping gene β-actin and represented relative to the WT cell line. Results with the control cell line did not differ from those obtained with the WT cell line.

**Figure 3 ijms-25-11512-f003:**
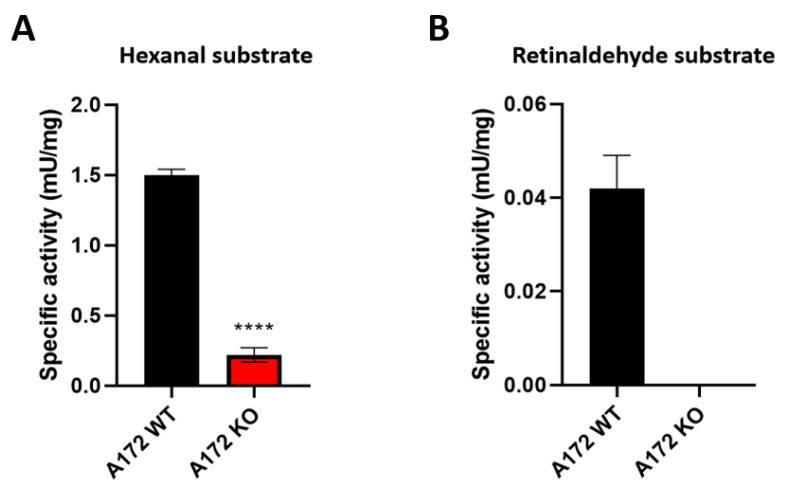
Comparison of ALDH activity in cellular extracts of A172 WT and A172 KO. (**A**) Activity using hexanal as substrate; (**B**) Activity using all-*trans*-retinaldehyde as substrate. Data are the mean ± SD of triplicates (n = 3). Specific activity is expressed in mU per mg of total protein present in the cellular extract; here, 1 mU is defined as 1 nmol of product generated per min. Asterisks indicate statistically significant difference, as analyzed by unpaired *t* test with Welch’s correction. **** *p* < 0.0001.

**Figure 4 ijms-25-11512-f004:**
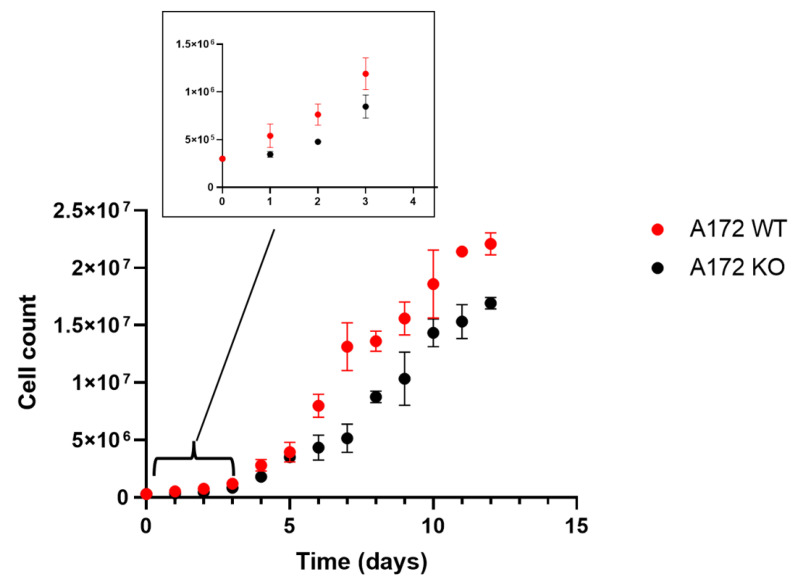
Growth curves for A172 WT and A172 KO cells. Inset: Zoom plot of days 1–4. The growth curves were analyzed with a general lineal model, and significant differences were found for main effects (day of culture, cell status) as well as the interaction between them. The increasing growth trend throughout the days was maintained, although at a notably lower rate. Bilateral comparisons (Student’s *t*-test) were also performed for cell counting at each day, showing values either significant (*p* < 0.05) or with *p* values between 0.05 and 0.1 for most comparisons. Overall, this outcome reflects a consistent impact on the cell growth rate. Data are the mean ± SD of triplicates (n = 3).

**Figure 5 ijms-25-11512-f005:**
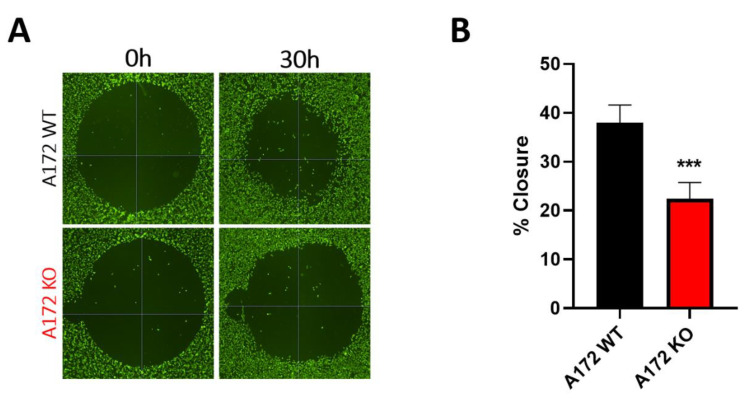
Comparison of migration capacity in A172 WT and A172 KO cells. (**A**) Fluorescence images showing migration areas of A172 WT and A172 KO cells after incubation with calcein dye, obtained in a SpectraMax plate reader at times 0 and 30 h; (**B**) percent closure of migration areas after 30 h in A172 WT and A172 KO cells. Data are presented as the mean ± SD of four replicates (n = 4). Statistical significance was determined using an unpaired *t* test with Welch’s correction, with asterisks indicating significant differences: *** 0.0001 < *p* < 0.001.

**Figure 6 ijms-25-11512-f006:**
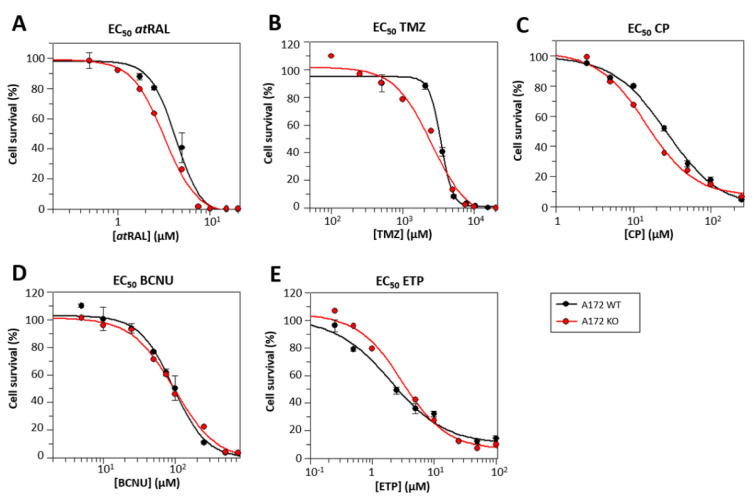
Dose–response curves for all-*trans*-retinaldehyde (atRAL) and FDA-approved drugs in A172 WT and A172 KO after 48 h of treatment. (**A**) atRAL; (**B**) TMZ; (**C**) CP; (**D**) BCNU; (**E**) ETP. Data points are the mean ± SE of triplicates (n = 3) in a single representative experiment, performed at least twice independently. Curves were obtained using GraFit 5.0 (Erithacus software). EC_50_ values for the studied compounds were significantly different (*p* < 0.05) when comparing A172 WT and A172 KO cell lines, with the exception of BCNU.

**Figure 7 ijms-25-11512-f007:**
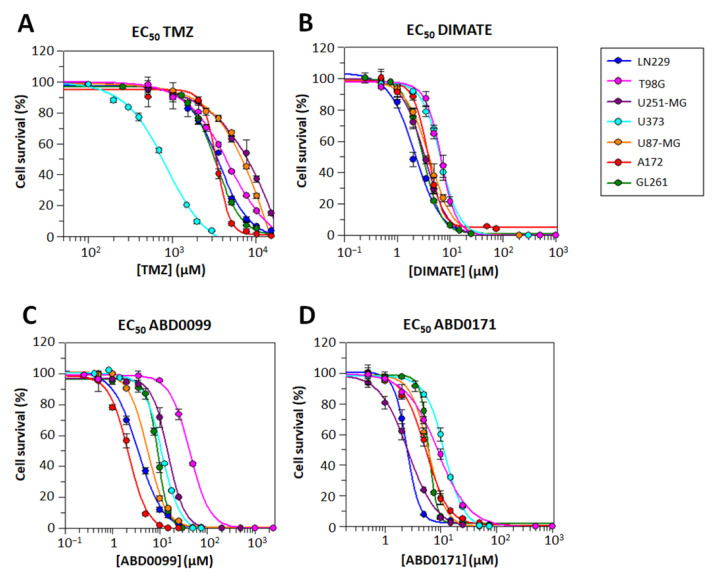
Dose–response curves for TMZ and ALDH inhibitors in the panel of GB cell lines after 48 h of treatment. (**A**) TMZ; (**B**) DIMATE; (**C**) ABD0099; (**D**) ABD0171. Data points are the mean ± SE of triplicates (n = 3) in a single representative experiment, performed at least twice independently. Curves were obtained using GraFit 5.0 (Erithacus software).

**Figure 8 ijms-25-11512-f008:**
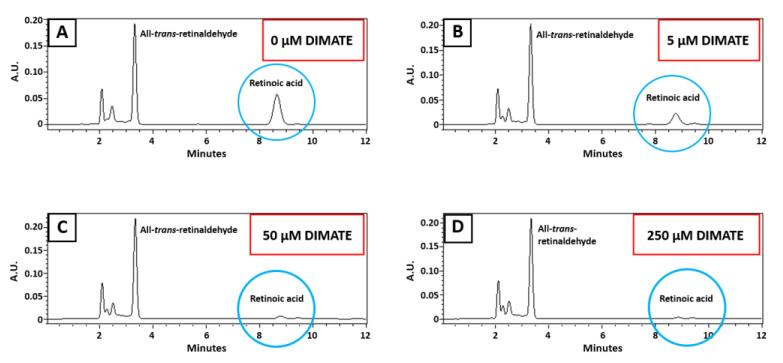
Representative HPLC analysis of ALDH activity in A172 cell extract using all-*trans*-retinaldehyde as substrate in the presence of increasing concentrations of DIMATE. (**A**) 0 µM DIMATE; (**B**) 5 µM DIMATE; (**C**) 50 µM DIMATE; (**D**) 250 µM DIMATE. Absorbance units (A.U.) are represented in the y axis, while retention time (minutes) is represented in the x axis. The peaks corresponding to the substrate, all-*trans*-retinaldehyde, and the reaction product, retinoic acid, (the latter circled in blue) appear at 3.5 and 8.5 min, respectively.

**Figure 9 ijms-25-11512-f009:**
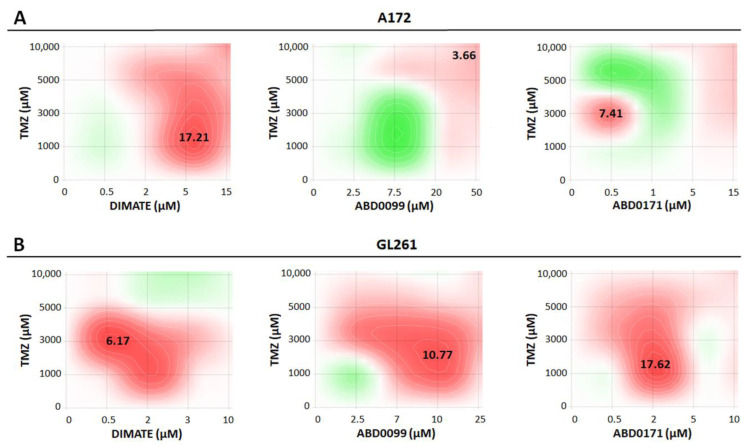
Synergy maps for 48 h treatments using TMZ combined with each ALDH inhibitor in (**A**) A172 and (**B**) GL261, obtained after analysis with SynergyFinder software (https://synergyfinder.fimm.fi/), according to the reference model of Bliss. The concentration of TMZ is represented in the y axis, whereas the concentration of each ALDH inhibitor is represented in the x axis. Synergy scores of the most synergistic areas are indicated in the map. Scores below −10 indicate antagonism (green); scores between −10 and 10 indicate an additive effect (light green to light red); scores above 10 indicate synergism (red). Shown here is the result of a single representative experiment consisting of duplicates (n = 2) for each drug concentration. The experiment was performed at least twice independently.

**Figure 10 ijms-25-11512-f010:**
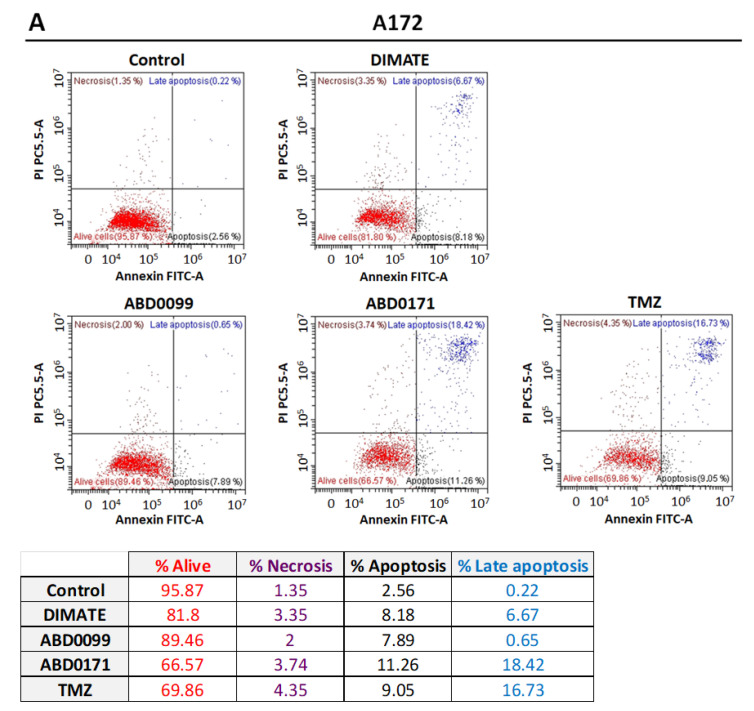
Cell death analysis by flow cytometry after 24 h of incubation with TMZ equimolar concentrations of DIMATE, ABD0099 and ABD0171 in (**A**) A172 and (**B**) GL261 cells. Iodide propidium labeling (necrosis) is represented in the y axis, whereas annexin V labeling (apoptosis) is represented in the x axis. Cells in the PI positive region of the cytogram could come from either necrotic or late-apoptotic cells. The result of a single representative experiment is shown here, consisting of duplicates (n = 2) for each drug treatment (to simplify, only one of the duplicates is shown). The experiment was performed at least twice independently.

**Figure 11 ijms-25-11512-f011:**
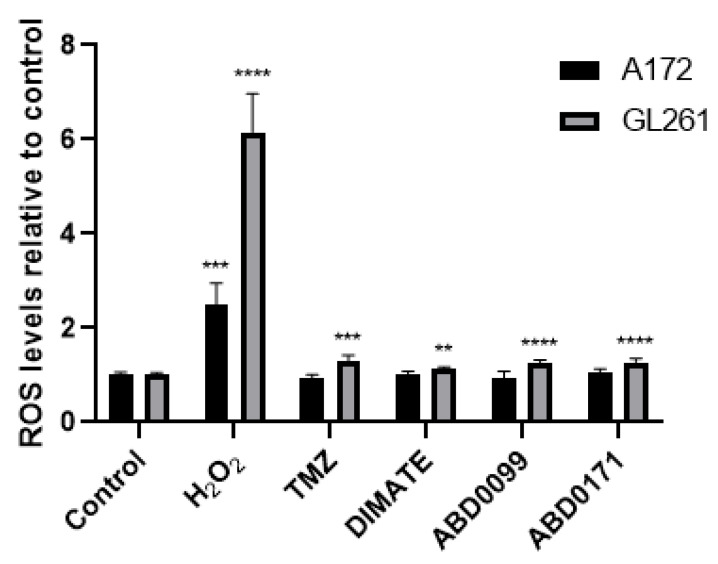
Relative intracellular levels of ROS in A172 and GL261 after 6 h of treatment with 500 µM H_2_O_2_ (used as a positive control), and TMZ or ALDH inhibitors at the EC_50_ concentration. Data are the mean ± SD of six replicates (n = 6) of a single representative experiment. The experiment was performed at least twice independently. Asterisks indicate statistical significance compared to the respective control, as analyzed by an unpaired *t* test with Welch’s correction. ** 0.001 < *p* < 0.01; *** 0.0001 < *p* < 0.001; **** *p* < 0.0001.

**Figure 12 ijms-25-11512-f012:**
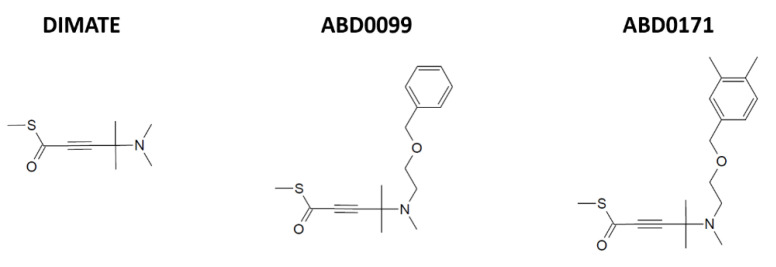
Chemical structures of the ALDH inhibitors tested in this study.

**Table 1 ijms-25-11512-t001:** Absolute values of ALDH specific activity (mU/mg) in A172 and GL261 cell extracts using hexanal or all-*trans*-retinaldehyde as a substrate. Specific activity is expressed in mU per mg of total protein present in the cell extract; here, 1 mU is defined as 1 nmol of product generated per min. Values are the mean ± SD of triplicates (n = 3).

	Substrate	
Cell Line	Hexanal	All-*Trans*-Retinaldehyde
A172	1.79 ± 0.02	0.05 ± 0.01
GL261	0.47 ± 0.06	ND

ND: not detected.

**Table 2 ijms-25-11512-t002:** EC_50_ values (µM) ± SE of all-*trans*-retinaldehyde (atRAL) and FDA-approved anti-cancer drugs in A172 WT and A172 KO cell lines after 48 h of incubation. Data represents the results from a single representative experiment, conducted in triplicates (n = 3) and repeated independently at least twice, analyzed using a 4-parameter equation with GraFit 5.0 (Erithacus Software, East Grinstead, U.K.).

	Compound				
Cell Line	atRAL	TMZ	CP	BCNU	ETP
A172 WT	4.25 ± 0.31	3390 ± 118	26.66 ± 2.89	93.08 ± 6.93	2.01 ± 0.39
A172 KO	3.20 ± 0.19	2573 ± 223	14.62 ± 1.65	96.54 ± 8.33	3.04 ± 0.55

**Table 3 ijms-25-11512-t003:** EC_50_ values (µM) ± SE of TMZ and ALDH inhibitors in the panel of GB cell lines after 48 h of treatment. The data represent the fitting of results from a single representative experiment with triplicates (n = 3), performed at least twice independently, to a 4-parameter equation using GraFit 5.0 (Erithacus software). The EC_50_ value is defined as the drug concentration at which half of the maximal response is achieved.

	Cell Line						
Drug	LN229	T98G	U251-MG	U373	U87-MG	A172	GL261
TMZ	3598 ± 248	4778 ± 490	>5000 *	855 ± 65	>5000 *	3390 ± 118	3281 ± 176
DIMATE	2.15 ± 0.27	6.72 ± 0.24	3.30 ± 0.08	6.62 ± 0.23	4.01 ± 0.15	3.77 ± 0.27	3.03 ± 0.08
ABD0099	3.41 ± 0.17	42.63 ± 1.11	14.92 ± 0.48	10.99 ± 0.40	5.39 ± 0.32	2.15 ± 0.13	8.95 ± 0.19
ABD0171	2.47 ± 0.08	8.83 ± 0.42	2.59 ± 0.11	11.78 ± 0.32	5.97 ± 0.20	5.34 ± 0.26	6.01 ± 0.09

* EC_50_ estimated value was greater than 5000 µM and could not be confidently determined due to the fact that 0% cell viability was not reached at the tested TMZ concentrations (higher concentrations were not feasible due to solubility issues).

**Table 4 ijms-25-11512-t004:** Remaining ALDH activities in A172 and GL261 cell extracts using hexanal as a substrate, in the presence of a 15 µM inhibitor. Values are the mean ± SD of triplicates (n = 3).

Cell Line	DIMATE	ABD0099	ABD0171
A172	68.51 ± 4.85%	93.32 ± 1.10%	39.75 ± 3.07%
GL261	71.22 ± 13.57%	68.15 ± 8.84%	50.61 ± 6.90%

**Table 5 ijms-25-11512-t005:** Remaining ALDH activities in A172 cell extract using all-*trans*-retinaldehyde as a substrate, in the presence of increasing concentrations of inhibitors. Values are the mean ± SD of duplicates (n = 2).

Inhibitor								
DIMATE			ABD0099			ABD0171		
5 µM	50 µM	250 µM	5 µM	50 µM	250 µM	5 µM	50 µM	250 µM
42.09 ± 4.64%	11.34 ± 1.36%	6.04 ± 1.03%	96.64 ± 2.29%	73.89 ± 1.50%	34.09 ± 0.09%	66.73 ± 1.45%	26.17 ± 1.40%	23.91 ± 2.29%

**Table 6 ijms-25-11512-t006:** Description of antibodies used in the immunoassay.

Primary Antibody	Reference	Dilution	Specificity	Secondary Antibody
ALDH1A1	Proteintech, 15910-1-AP	1/5	Human, mouse	Anti-rabbit (WES detection kit)
ALDH1A2	Proteintech, 13951-1-AP	1/10	Human, mouse	Anti-rabbit (WES detection kit)
ALDH1A3	VWR, ABGEAP7847A	1/5	Human, mouse	Anti-rabbit (WES detection kit)
ALDH1B1	Proteintech, 15560-1-AP	1/50	Human, mouse	Anti-rabbit (WES detection kit)
ALDH2	Proteintech, 15310-1-AP	1/50	Human, mouse	Anti-rabbit (WES detection kit)
ALDH3A1	Proteintech, 15578-1-AP	1/50	Human, mouse	Anti-rabbit (WES detection kit)
ALDH3A2	Proteintech, 15090-1-AP	1/10	Human, mouse	Anti-rabbit (WES detection kit)
ALDH3B1	Abcam, ab236673	1/50	Human, mouse	Anti-rabbit (WES detection kit)
ALDH3B2	Proteintech, 15746-1-AP	1/50	Human, mouse	Anti-rabbit (WES detection kit)
ALDH7A1	Proteintech, 10368-1-AP	1/10	Human, mouse	Anti-rabbit (WES detection kit)
Loading control	ProteinSimple, 042-196	1/10	Rabbit	Anti-rabbit (WES detection kit)

**Table 7 ijms-25-11512-t007:** Description of the primers used for PCR.

Target Gene	Forward Primer (5′ to 3′)	Reverse Primer (5′ to 3′)
*ALDH1A1*	TTGGAAATCCTCTGACCCCA	CCTTCTTTCTTCCCACTCTC
*ALDH1A2*	CATTGGAGTGTGTGGACAGA	GGAGCTATTTTCCAGGCA
*ALDH1A3*	TTTTCATCGACCTGGAGG	GACGTTGTCATCTGTGGG
*ALDH1B1*	ACTTGGCCTCACTCGAGA	CCAGCAAAGTACCGATAC
*ALDH2*	GTCAGATGCCGATATGGAT	GCCCTGGTTGAAGAACAG
*ALDH3A1*	CACATCACCTTGCACTCTCT	AGCTCTTCTTGCCATGGT
*ALDH3A2*	TAGCTTTTGGTGGGGAGA	CTTGCATCACCTTGGTTT
*ALDH3B1*	TATCTAATCACGGGCCAC	AGCTGCTTGTTTTCTTGC
*ALDH3B2*	TTCTCCAACAGCAGCCAG	CGGACAGCAGAGATATGTAG
*ALDH7A1*	GACCTATTGCCCTGCTAA	CCATGCTTCTCTTGCTTTC

## Data Availability

Data will be shared upon reasonable request to corresponding authors.

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
