# Peer review of "Targeting Retinaldehyde Dehydrogenases to Enhance Temozolomide Therapy in Glioblastoma"

_ijms, 2024, doi:10.3390/ijms252111512_

Round 1
Reviewer 1 Report
Comments and Suggestions for Authors
-
Title: Targeting retinaldehyde dehydrogenases in combined therapy against glioblastoma – The title clearly conveys the subject of the study, though it might benefit from specifying which therapeutic strategies are combined.
-
Abstract:
- Strengths: The abstract concisely describes the issue of glioblastoma (GB) chemoresistance, specifically related to temozolomide (TMZ), and the role of aldehyde dehydrogenases (ALDHs) in this resistance.
- Areas for improvement:
- The phrase "proved to be ALDH inhibitors" could be reworded to "demonstrated efficacy as ALDH inhibitors."
- "Our results strongly suggest" is a bit vague; it might be useful to briefly state key findings or percentages to support this claim.
-
Introduction:
- The introduction provides a comprehensive background on glioblastoma, ALDH's role, and the rationale for the study. It flows well and builds a logical case for the need to investigate ALDH inhibitors. However, more specific gaps in current knowledge could be emphasized to highlight what this paper adds to the field.
- When introducing the challenges of glioma care and the need for advanced surgical interventions, you can cite:
- Advancements in Glioma Care: Focus on Emerging Neurosurgical Techniques. Biomedicines. 2024;
- 12(1):8. https://doi.org/10.3390/biomedicines12010008
- Methodology:
-
- The methods are detailed, including Western blot analysis, cell viability assays, and ALDH activity assays. One suggestion is to ensure all statistical tests are explicitly mentioned with significance levels, which is partially done but could be more systematic.
-
Results:
- The results are well-presented with sufficient data, but a few points could be streamlined. Some graphs (e.g., Figure 3) might benefit from a clearer explanation of what the trends signify for readers unfamiliar with the technical aspects.
- The discussion on ALDH knockout effects is strong but could reference more recent studies, especially when discussing its implications for growth rates and drug sensitivity.
- When explaining the relevance of advanced imaging in studying tumor behavior or treatment response in fact you should cite: Clustering Functional Magnetic Resonance Imaging Time Series in Glioblastoma Characterization: A Review of the Evolution, Applications, and Potentials. Brain Sci. 2024, 14, 296.
- Figures and Tables:
-
- The figures are informative but might benefit from larger labels or annotations explaining key data points, especially for readers less familiar with biochemical assays.
-
Discussion:
- Strengths: The discussion ties the findings back to the broader context of glioblastoma treatment and drug resistance.
- Areas for improvement: Consider more critical reflection on the limitations of the study. For example, discussing the potential for in vivo applications based on these in vitro results could deepen the discussion.
Minor revision needed
Reviewer 2 Report
Comments and Suggestions for Authors
The current research article entitled “Targeting retinaldehyde dehydrogenases in combined therapy against glioblastoma” by Jimenez et at is focused on exploring the therapeutic options to treat glioblastoma using cell line model (in vitro) on metabolic axis. Authors have studied aldehyde dehydrogenases (ALDHs), an important functional gene to drive tumor progression. Authors have used ALDH inhibitor to overcome cancer cell proliferation. Authors have examined the combinatorial approach of ALDH inhibitors and temozolomide (TMZ), a standard care therapeutic in treatment of GBM. Authors have performed many biochemical assays to evaluation of effect of combination treatment of ALDH inhibitors and TMZ in vitro.
Specific comments:
ALDHs (including. subfamilies) promoter regulation depends on multiple transcription factors and enhancers, reported in multiple cancer types including GBM. Did authors have any findings on this axis?
Lone 149-151 “GL261 was chosen….”, is this the limitation of the study?
The drugs sensitivity has no significant differences between WT and KO cell lines. Explanation, I would suggest authors to deep dive to understand this.
I wonder, if authors have performed only bliss score for combination treatment. I would suggest authors to perform more experiment in combination. Moreover, I would be better if authors can overcome this issue by using TMZ0- resistant cells to explore ALDH inhibitors, which may give more exciting approach to understand why we need new therapeutic option for standard care in GBM.
Authors have also performed experiment to understand ROS level after different treatments in cell lines. Since, ROS have been reported in multiple aspects to inhibit cell proliferation. I would suggest authors to perform additional experiment.
Minor comments:
Identifiers for all the cell lines used in study
Appropriate Statistical significance and n =? in each experiment (figure legends)
Since, many experiments can be placed in same figure with by making different panels, I would suggest author to make comprehensive figure by quality, not by enumerative. (for eg, figure 3, 4 and 5)
Add limitation for the study (stating only ion cell line were explored in most of the experiments)
Typos, if any.
Comments on the Quality of English LanguageI would recommend using professional English editing services to enhance the clarity and readability of the manuscript.
Reviewer 3 Report
Comments and Suggestions for Authors
The manuscript entitled „Targeting retinaldehyde dehydrogenases in combined therapy against glioblastoma” is interesting and touches vital topic of ALDH-related chemoresistance in glioma stem cells.
In the introduction authors placed more than 50 references thus adequately introducing the readers to the scientific area. Justification of the project is satisfactory.
In the manuscript authors aim to investigate three novel ALD inhibitors (namely DIMATE 108 and its analogues, ABD0099 and ABD0171) on in vitro cultured seven GB cell lines, including six human cell lines (LN229, T98G, U251-MG, 122 U373, U87-MG and A172) and one murine cell line (GL261), and additionally non-cancerous human astrocytic cell line derived from the stem cell line Ax-0019.
However, the reviewer has some remarks and questions which need to be answered.
In the introduction, the authors frequently refer to glioma stem cells (GSC), however, there is no follow up of this idea in the manuscript – please explain.
Line 138: what authors mean by “subpopulations of GSCs,” and the citation is needed.
Line 150: justify this statement: “being a suitable line to be used in immunocompetent models for future in vivo preclinical studies”
Given the various expression of ALD isoforms in the cell lines, it would be rational to perform ALDH activity tests on cell lines expressing only ALDH1A3 isoform (i.e. U373) to find out to what extent this isoform is important in overall aldehyde dehydrogenic activity.
Line 187: Please explain and discuss how “the most affected isoforms were the enzymes belonging to ALDH1 family” after ALDH1A3 isoform knockout – was the knockout nonspecific? – refer to the lack of negative control in this experiment.
Figure 1. A graphical representation showing the differences of the expression levels between the cell lines is needed
Figure 2. the information on how the mRNA expression was calculated is missing
Table 2 is erroneous since it does not present A172 WT and A172 KO cell lines.
Table 3: what is the cytotoxicity of the inhibitors towards Ax-0019 cell line?
Figure 10 is hard to read – a table would help.
Line 562: rephrase and correct
Chapter 4.8. Please provide more details on the selection and sorting of the cells/clones. How many clones were screened? Were there any negative/positive controls used in the study?
Comments on the Quality of English LanguageRephrase and correct information in materials and methods.
Round 2
Reviewer 1 Report
Comments and Suggestions for Authors
Dear Authors,
When explaining the relevance of advanced imaging in studying tumor behavior or response to treatment, you should cite: Clustering Functional Magnetic Resonance Imaging Time Series in the Characterization of Glioblastoma: A Review of the Evolution, Applications, and Potential. Brain Sci. 2024, 14, 296
Reviewer 2 Report
Comments and Suggestions for Authors
The authors have adequately addressed my concerns.
Reviewer 3 Report
Comments and Suggestions for Authors
I am satisfied with the Authors' response and implemented corrections